# SGC-VSLAM: A Semantic and Geometric Constraints VSLAM for Dynamic Indoor Environments

**DOI:** 10.3390/s20082432

**Published:** 2020-04-24

**Authors:** Shiqiang Yang, Guohao Fan, Lele Bai, Cheng Zhao, Dexin Li

**Affiliations:** School of Mechanical and Precision Instrument Engineering, Xi’an University of Technology, Xi’an 710048, China; 2170220051@stu.xaut.edu.cn (G.F.); 2180220045@stu.xaut.edu.cn (L.B.); 2190221126@stu.xaut.edu.cn (C.Z.); lidexin@xaut.edu.cn (D.L.)

**Keywords:** Visual SLAM, ORB-SLAM2, dynamic indoor environment, dynamic feature filtering, point cloud map

## Abstract

As one of the core technologies for autonomous mobile robots, Visual Simultaneous Localization and Mapping (VSLAM) has been widely researched in recent years. However, most state-of-the-art VSLAM adopts a strong scene rigidity assumption for analytical convenience, which limits the utility of these algorithms for real-world environments with independent dynamic objects. Hence, this paper presents a semantic and geometric constraints VSLAM (SGC-VSLAM), which is built on the RGB-D mode of ORB-SLAM2 with the addition of dynamic detection and static point cloud map construction modules. In detail, a novel improved quadtree-based method was adopted for SGC-VSLAM to enhance the performance of the feature extractor in ORB-SLAM (Oriented FAST and Rotated BRIEF-SLAM). Moreover, a new dynamic feature detection method called semantic and geometric constraints was proposed, which provided a robust and fast way to filter dynamic features. The semantic bounding box generated by YOLO v3 (You Only Look Once, v3) was used to calculate a more accurate fundamental matrix between adjacent frames, which was then used to filter all of the truly dynamic features. Finally, a static point cloud was estimated by using a new drawing key frame selection strategy. Experiments on the public TUM RGB-D (Red-Green-Blue Depth) dataset were conducted to evaluate the proposed approach. This evaluation revealed that the proposed SGC-VSLAM can effectively improve the positioning accuracy of the ORB-SLAM2 system in high-dynamic scenarios and was also able to build a map with the static parts of the real environment, which has long-term application value for autonomous mobile robots.

## 1. Introduction

Simultaneous Localization and Mapping (SLAM), a prerequisite for many robotic applications, involves a system that simultaneously completes the positioning of the mobile robot itself and the map construction of the surrounding environment without any prior environmental information [1,2]. Visual SLAM (VSLAM), where the primary sensor is a camera, has received increasingly more attention in recent years and can be classified into two methods: Feature-based methods [3,4,5] and direct methods [6,7]. The feature-based method extracts the features in each frame to estimate a self-pose that has better environmental adaptability, whereas direct methods estimate the pose by adopting the minimization of photometric errors, which are more sensitive to light changes compared to feature-based methods.

Researchers have already addressed SLAM from many different angles. However, some problems in VSLAM have not been adequately solved until now [8,9]. For example, the standard ORB (Oriented FAST and Rotated BRIEF) algorithm tends to reserve the feature points at the strong texture regions [10]. As a consequence, when dynamic objects have strong texture information, a large number of features will be extracted from the dynamic objects, which will introduce errors to the SLAM system. Besides, the feature points concentrated in the local regions cannot well reflect the information of the entire image. When the concentration areas of the feature points in two matching frames are different, the number of feature matches will largely reduce, resulting in tracking failure. Therefore, the problem that the feature points are easily concentrated in the local area needs to be solved.

In recent research, the vast majority of SLAM systems adopt a static scenario assumption for convenience analysis; therefore, the static computation models of these systems may be impacted by such dynamic objects [11,12]. In detail, the dynamic features are unavoidably used for computing the robot poses, which introduces errors to the visual odometer. Moreover, when a robot returns to the previously arrived scene where the dynamic targets have gone away, the loop detection would be confused by matching the same scene but with the different map point. Thus, eliminating the negative impact of dynamic objects on SLAM systems is a critical challenge for VSLAM.

Map construction is core work for VSLAM system. However, building a high-quality environment map is hard work in highly dynamic environments. If the dynamic objects in the scene are not handled properly, moving objects will be recorded in the resulting map, which makes the map virtually unable to be used for future applications, such as the map-based navigation. To improve the stability and robustness of SLAM system in dynamic indoor environments, this paper proposes a semantically and geometrically constrained VSLAM, i.e., SGC-VSLAM, which adopts a novel deep learning and geometric combination to filter out outliers and, at the same time, generate a static point cloud map. The main contributions of this paper are summarized as follows:(1)An improved quadtree-based ORB algorithm, which limits the maximum iteration depth and extracts features by an adaptive method, was proposed to extract stable ORB features and solve the problem that ORB feature extractor is sensitive to illumination changes.(2)A semantic and geometric constraints’ algorithm (SGC) was proposed to filter out dynamic features. SGC modified epipolar constraint by fusing semantic information, which can effectively reduce the influence of dynamic objects.(3)A novel drawing key frame selection strategy was designed based on map point information to reduce redundant environmental information, and a static point cloud map was constructed by removal of the point cloud generated from the dynamic objects.

The main content of this paper is arranged in the following sections: Section 2 reviews the related work. Section 3 introduces the proposed improved SLAM algorithm system’s framework and its detailed process. Section 4 tests the proposed method experimentally, and the conclusions and future work are described in the last part of the paper.

## 2. Related Work

Despite many obstacles, there remains motivation for finding a novel VSLAM algorithm for indoor dynamic scenes. Because the ORB features are prone to concentrate in the strong texture region, the quadtree-based method was adopted by ORB-SLAM2 to extract uniform ORB features [5]. However, the feature points extracted by quadtree-based can easily yield an over-uniform distribution that reserves some weak quality feature points, and a large number of iterations affects its running efficiency. A Uniform Robust-SIFT (UR-SIFT) algorithm was proposed by Sedaghat et al. [13], which can adaptively control the number of SIFT feature points in different regions. This is a better algorithm for uniformly distributing SIFT feature points. Paul and Pati [14] presented a modified UR-SIFT, which effectively generates robust, reliable, and uniformly distributed aligned feature points. So far, most of the uniform feature distribution algorithms proposed by scholars have shown high computational complexity, and their matching effects were not ideal.

The core challenge to improve the robustness of a SLAM system in dynamic environments is detecting and filtering dynamic features in the scenario [15], which means preventing the visual odometers from using the features extracted from dynamic objects. There are many excellent classical dynamic detection methods. However, the main problem in SLAM is the movement of the camera. Here, classic motion segmentation methods, such as background subtraction and the frame difference method, are useless because the foreground and background move independently in the image [16]. Moving target detection under a dynamic background has become a new challenge. Wangsiripitak and Murray [17] avoided moving outliers by tracking known 3D dynamic objects. The work of Moo et al. [18] used two single Gaussian models that can effectively represent the foreground and the background. Sun et al. [19] extended the concept of using an intensity difference image to identify the boundaries of dynamic objects. However, the large amount of required computation hinders the real-time applicability of this model. Kim et al. [20] proposed to obtain the static parts of a scene by computing the difference between consecutive depths of images projected over the same plane. Similarly, Zhao et al. [21] used depth images to detect dynamic objects. However, these methods were prone to be affected by the uncertainty of depth images. More recently, Li and Lee [22] used depth edge points with associated weights indicating their probability of belonging to a dynamic object. The above methods all solely use multiview geometry to detect dynamic objects. With the rapid development of deep learning methods, some tasks that were considered impossible in the past can be done, such as target detection.

Advanced convolutional neural network (CNN) architectures like YOLO [23], SSD (Single Shot multibox Detector) [24], SegNet [25], and Mask R-CNN [26] are considered state of the art and can effectively detect the labels of the objects in a scene, which is also accomplished by the SLAM system. Zhang et al. [27] used YOLO running in a dependent thread to detect semantic information; they considered the features extracted from moving objects as unstable and filtered them out. Similarly, Li et al. [28] also used YOLO to detect dynamic features and proposed a novel sliding window compensation algorithm to reduce the detection errors of YOLO, thereby providing a new idea for detecting dynamic objects. Zhong et al. [29] used an SSD network to detect objects labelled as dynamic and then filtered out all the features extracted from it. Some recent works have combined the dynamic detection results from multiview geometry and deep learning methods. Berta Bescos et al. [15] added a dynamic target detection module on ORB-SLAM2, which can accurately identify and segment dynamic targets through a combination of multiview geometry and Mask R-CNN. Features that are either semantically dynamic or geometrically dynamic are all considered as “dynamic”. Similarly, Chao Yu et al. [30] filtered the feature points extracted on a dynamic target using the epipolar constraint and the SegNet network. However, the fundamental matrix used for the epipolar constraint has a significant error, which affects the performance of the algorithm. More recently, Han and Xi [31] proposed a PSPNet-SLAM (Pyramid Scene Parsing Network-SLAM) to improve ORB-SLAM2, where PSPNet and the optical flow were used to detect dynamic features. The features extracted from objects labelled as dynamic and features with large optical flow values were all filtered out, and the remaining features were used for tracking. This method achieved high positioning accuracy in the TUM dataset. The above dynamic detection methods retain some problems, such as not working in real time and the inability to combine the deep learning method and the multiview geometry method well to detect dynamic targets. There are many excellent methods to construct a static map in a dynamic environment. DS-SLAM [30] can generate an octotree map, and DynaSLAM [15], using the background inpainting method, can create a dense static map. These methods all use semantic information to prevent the dynamic points that appear in a map. However, there are few works on how to select the frames to construct such a map.

## 3. Improved System Description

In this section, SGC-VSLAM will be introduced from four main elements. First, the overall-architecture of SGC-VSLAM is presented. Second, the improved quadtree-based algorithm that is used to make features more evenly distributed is introduced. Subsequently, the dynamic feature detection method is introduced, which combines semantic segmentation and a geometric constraint to filter out dynamic features. Finally, the method for building a static point cloud map is presented.

### 3.1. Improved System Framework Overview

An overview of the proposed SGC-VSLAM system is shown in Figure 1. This overview can be divided into five main threads: Tracking, local mapping, loop closing, semantic detection, and point cloud map creation. Among these threads, the semantic detection and point cloud map creation threads are two new threads added to ORB-SLAM2. In the tracking thread, an improved quadtree-based algorithm was used to extract stable and uniform ORB feature points for the current frame image. Then, the dynamic feature points extracted on the moving target are filtered by the proposed semantic and geometric constraints’ method. Only the static feature points are input into the tracking thread for pose estimation. Finally, a static point cloud map is generated by the point cloud map creation thread. The local mapping and loop closing threads are the same as those of ORB-SLAM2 [5].

### 3.2. Improved Quadtree-Based ORB Feature Uniform Distribution Algorithm

Extracting features is the core work of the feature-based VSLAM. ORB-SLAM2 uses the quadtree-based algorithm to extract uniform ORB features, thus avoiding the problem of the features gathering together. However, some potential problems may still be ignored. In this section, the defects of the standard quadtree-based algorithm are discussed in detail, and then the improved quadtree-based algorithm is proposed to enhance the standard algorithm.

#### 3.2.1. Analysis of the Standard Quadtree-Based Algorithm

A schematic diagram of the quadtree-based algorithm is shown in Figure 2. For example, the image with 12 ORB features is first segmented as Quadtree nodes, and then the features in the node are reserved based on their Harris response values. Figure 2b presents the results of the final reserved ORB features of the standard quadtree-based algorithm. The detailed flow of the standard quadtree-based algorithm is as follows:
Step 1: Initialize the root node, including all the features, which will serve as the parent node for the next segmentation.Step 2: Segment all parent nodes with quadtree nodes and remove the child nodes that have no features.Step 3: If the child node with only one feature or the total numbers of the node are more than the desired features, move to Step 4. Otherwise, move to Step 2.Step 4: Reserve the features with the highest Harris response values in each child node as the ultimate ORB features.

The above flow shows that the Step 3 is the quit condition for the quadtree node segmentation, which determines the segment depth as well as the uniformity of features. This node segmentation method may cause some problems under a high-dynamic environment. For example, the grey area in Figure 2a represents the motion blur region, in which the grey difference between pixels is small. The Harris response value is a measure of the grey difference between the feature and its surrounding pixels; the more significant the grey difference, the higher the response value [32]. Therefore, the features extracted in the motion blur region have a low Harris response value, which is not conducive to subsequent matching and tracking. The node segmentation result when setting the number of the desired features to 9 is shown in Figure 2a, and the final reserved features of the standard quadtree-based algorithm are shown in Figure 2b. It can be seen that features 9 and 10 are finally reserved primarily due to the excessive node segmentation. The above example only mentions the problems caused by dynamic objects. The features extracted from regions with weak textures, such as desktops, floors, screens, etc., will also have small grey differences. These features will also be reserved by the standard quadtree-based algorithm for the same reason. Further, whenever the parent node is segmented, the algorithm has to calculate all the features in the parent node belonging to that child node. Hence, over-segmentation will also impact the running efficiency.

Based on the above analyses, two defects significantly impact the performance of the standard quadtree-based algorithm: Over-uniform distributed ORB features and long iteration times. An improved quadtree-based algorithm was introduced to solve the above two defects.

#### 3.2.2. Improved Quadtree-Based Algorithm

To reasonably distribute the feature points to be extracted, the feature extraction number of each layer of the image pyramid is set as follows:(1)$$DesF_{i}=N\times \frac{1-InvSF}{1-InvSF^{n}}\times InvSF^{i}$$
where the total desired number of ORB feature points is denoted as N, n represents the total number of the layers of the pyramid, $DesF_{i}$ is the number of feature points required for the i layer, and InvSF represents the reciprocal of the scale factor.

FAST (Features from Accelerated Segment Test) features [33] are extracted when the feature extraction number of each layer in the image pyramid is determined. In ORB-SLAM2, the extraction threshold of FAST is set according to the engineering experience value, which does not agree with real environments. For example, the high threshold cannot extract enough features in the image regions where the texture information is weak. Hence, a threshold that can adapt to image changes was set in this method, and the initial threshold *iniTH* was set as
(2)$$iniTh=\frac{1}{\kappa \cdot ni}\sum_{x=1}^{ni}\left(I(x\right)-\kappa {)}^{2}$$
where I(x) is the grey value of each pixel in the image, and κ is the average value of the image in grey. The total number of image pixels is represented as *ni*.

Next, every quadtree node is segmented on the image. Unlike the standard quadtree algorithm, the number of the divided nodes are limited in the improved algorithm. The number of nodes depends on the depth of quadtree division and their relationship can be expressed as:(3)$$CNodes=4^{d}$$
where *d* represents the depth of quadtree division and CNodes is the number of nodes under the current depth. Thereby, the depth of quadtree division is used to control the number of nodes in the improved algorithm. In detail, if the current depth *d* is less than the maximum depth, then continue to divide the quadtree node; otherwise, stop the node division.

The most challenging task of the improved algorithm is the last step which is the selection of the final features of every child node. The following is the feature retention strategy. First, if a node has only one feature point, that point is reserved. Then, for the nodes with more than two feature points, all the feature points are sorted according to their Harris response values, thereby reserving the feature with the highest Harris response value in each node and filtering that feature in its node. This process is repeated until the number of reserved features is satisfied with the desired numbers. Besides, the minimum Harris response value is added in the feature retention strategy. Thus, if the Harris response value of the desired feature to extract is lower than the minimum threshold, then the features in this node are no longer extracted in the next iteration. Figure 2c shows a feature extraction schematic diagram of the improved algorithm, and the segmentation iteration depth was set as two. It can be seen that the dynamic features were no longer reserved by the improved algorithm because, among the Harris response values of feature 6, 7 were higher than those of the features in the motion blur region. The detailed node division and feature retention strategy flow is shown in Algorithm 1.
**Algorithm 1.** Improved quadtree-based feature points’ uniform distribution algorithmInput: image sequenceOutput: saved ORB feature points *Kp*1: Build image pyramid2: Extract Fast features by adaptive threshold3: for every pyramid layer *l* do4:   divide the quadtree node on the *l* layer pyramid5:   while *d*←0 < *maxD* do6:    for every child node *i* do7:     if *nNum*(*i*) > 1 && !*noFlag*(*i*) do8:       divide the child quadtree node in the parent *i* node9:     else10:      *noFlag*(*i*) = 111:      end if12:     end for13:     d++14:   end while15:   Sort features in every divided node from large to small according to Harris response value16:   while *Kp* < *DesF_i_* do17:     for every node *i* do18:      if *nNum*(*i*) > 1&& *maxkp* > *minH* do19:       add *maxkp* to *Kp* and erase it in the *i* node20:      end if21:     end for22:    end while23: end for
where *maxD* is the iteration maximum depth, *nNum*(*i*) represents the number of the feature points in the *i* node, the feature points with the maximum Harris response value in the node are denoted as *maxkp*, and *minH* is the minimum threshold of the Harris response value that can be accepted.

### 3.3. Semantic and Geometric Constraints’ Algorithm

A novel dynamic ORB feature-filtering approach called the semantic and geometric constraints’ algorithm (SGC) is introduced in this section, and the flowchart for the SGC algorithm is shown in Figure 3. First, every ORB feature extracted by the improved quadtree-based algorithm is filtered by the semantic bounding box generated from YOLO v3. Then, the rest of the features are matched with the previous frame based on the optical flow approach [34], and the matched feature pairs are then used to calculate the stable fundamental matrix. Finally, all the truly dynamic features are filtered by the geometric constraint.

#### 3.3.1. Semantic Detection

SGC-VSLAM adopts YOLO v3 to detect the objects labelled as dynamic. YOLO v3 can be implemented in real time on an embedded platform. Trained on PASCAL VOC (Visual Object Classes) dataset [35], this method can detect 20 classes in total. In an indoor environment, people are most likely to be dynamic objects, so people are labelled as dynamic objects in SGC-VSLAM.

Filtering out all of the features extracted from objects labelled as dynamic is a simple way to boost the accuracy of the SLAM system. However, there are two problems in this method. First, people are not the only independent dynamic objects in a real indoor environment. For example, a chair moved by people is also a dynamic object. Second, YOLO v3 can only provide a detection box, which may contain many static features. When the detection box occupies more than half of the current frame, the rest of the features are few, which may cause failure of the tracking thread in the SLAM system. In summary, using only semantic information filtering for all of the features extracted from the objects labelled as dynamic is insufficient to improve positioning accuracy and will cause tracking failures in some extreme cases.

#### 3.3.2. Geometric Constraint Method

Given the limitations of semantic constraints, SGC-VSLAM uses a geometric constraint method to filter out dynamic features further. The epipolar constraint, which is a part of photographic geometry, exists independent of the external environment between the two images and only depends on the camera’s internal parameters and relative poses. The relationship of the epipolar constraint between two adjacent images is shown in Figure 4.

In Figure 4, $I_{1}$ and $I_{2}$ are two images captured from the same scene. *P* is a spatial point whose projections on $I_{1}$ and $I_{2}$ are $x_{1}$ and *x_2,_* respectively. *O*_1_ and *O*_2_ are the optical centers of the two cameras, and their connections intersect $I_{1}$ and $I_{2}$ at $e_{1}$ and $e_{2}$, respectively. The $l_{1}$ and $l_{2}$ are the polar line of $I_{1}$ and $I_{2}$, which can be obtained by intersecting the $I_{1}$, $I_{2}$, and $PO_{1}O_{2}$ image planes.

If the spatial geometric transformation between $I_{1}$ and $I_{2}$ is noted as (*R*,*t*), then $x_{1}$ and *x_2,_* can be expressed as
(4)$$\begin{array}{l} s_{1}x_{1}=kp_{1}\\ s_{2}x_{2}=kp_{2}=k(Rp_{1}+t) \end{array}$$
where $p_{1}$,$p_{2}$ are the coordinates of *P* under the $I_{1}$ and $I_{2}$ camera coordinate systems, $s_{1}$,$s_{2}$ are their depth values, and *k* is the parameter matrices of the camera.

If the normalized coordinate $\bar{p_{1}}=k_{}^{-1}x_{1},\bar{p_{2}}=k_{}^{-1}x_{2}$, then we can obtain the following equation:(5)$$\bar{p_{2}}=R\bar{p_{1}}+t.$$

Multiply the above equation by ${\left(\bar{p_{2}}\right)}^{T}{\left[t\right]}_{x}$ and we obtain
(6)$${\left(\bar{p_{2}}\right)}^{T}{\left[t\right]}_{x}R\bar{p_{1}}=0$$
where ${\left[t\right]}_{x}$ is the antisymmetric matrix of the vector *t*. Substitute Equation (4) into Equation (6) and we obtain the following equation:(7)$$\begin{array}{l} x_{2}{}^{T}Fx_{1}=0\\ F=k_{}^{-T}{\left[t\right]}_{x}Rk_{}^{-1} \end{array}$$
where *F* is the fundamental matrix. The above equation illustrates the correct relationship of a matched feature pair. In other words, if a feature point with another feature point is a correct match, it must be on the polar line of its image plane. However, in a real scene, the calculated fundamental matrix often has some errors due to the influence of noise and camera calibration parameter errors, often making the feature points unable to fall exactly on the calculated polar lines. Therefore, the distance of each feature point is used to measure the reliability of the matching point pair.

If the homogeneous coordinates of $x_{1}$ and $x_{2}$ are $x_{1}=[u_{1},v_{1},1],x_{2}=[u_{2},v_{2},1]$, respectively, then the epipolar line on *l_1_* can be calculated by the following equation:(8)$$l_{2}=\left[\begin{array}{l} X\\ Y\\ Z \end{array}\right]=Fx_{1}=F\left[\begin{array}{l} u_{1}\\ v_{1}\\ 1 \end{array}\right]$$
where *X, Y,* and *Z* represent the epipolar equation vector. Then, the distance from point *x*_2_ to the epipolar line can be obtained by the following formula:(9)$$D=\frac{\left|x_{2}^{T}Fx_{1}\right|}{\sqrt{{\left\|X\right\|}^{2}+{\left\|Y\right\|}^{2}}}$$
where *D* represents the distance from *x*_2_ to the epipolar line. In a real scene, there are generally two situations that cause this distance to be too large. The first is that the feature points do not match, and the second is that the feature points are extracted from a dynamic target. Using a constraint on distance *D* can remove both the mismatched point pairs and the dynamic features. If D>θ, this point pair is regarded as a dynamic feature, where θ is the preset threshold value, and it can be calculated from Equation (10)
(10)$$\theta =\frac{\sum_{i=0}^{N}e^{-D_{i}}}{N}$$
where the total desired number of ORB feature points is denoted as N and $D_{i}$ is the distance from ith feature to its epipolar line.

#### 3.3.3. Dynamic Feature-Filtering Strategy

Using only semantic information cannot fully filter the dynamic features, which was discussed in the previous section. Like the semantic approach, only using a geometric constraint approach also has some problems. In detail, the calculated fundamental matrix will include large errors when only some dynamic features are used to calculate the fundamental matrix.

The SGC algorithm uses a combination of semantic and geometric constraints to identify the dynamic features of the scene. The flowchart of the SGC algorithm can be seen in Figure 3. First, every ORB feature extracted by the improved quadtree-based algorithm is filtered by the semantic bounding box generated by YOLO v3. Then, the rest of the features are matched with the previous frame based on the optical flow approach, and the matched feature pairs filtered by the bounding box of the objects labelled as dynamic are then used to calculate the stable fundamental matrix. Finally, all the truly dynamic features are filtered by the geometric constraint. The detailed dynamic feature-filtering flowchart can be seen in Algorithm 2:
**Algorithm 2.** Dynamic feature-filtering strategy Input: image sequenceOutput: static ORB featuresExtract ORB features and filter it using sematic bounding box generated by YOLOMatch the rest ORB features with the previous frame based on the optical flow approachCalculate the fundamental matrix between the two framesfor every matched feature pair *i* do if *i* is in the *Dyn* do  if $D<e_{1}\theta$ do   *i* append to *Kps*  end if else  if $D<e_{2}\theta$ do   *i* append to *Kps*  end ifend ifend for
where *Dyn* is the semantic bounding box regions of the objects labelled as dynamic. *D* denotes the distance between the features and the epipolar line, *Kps* is the final reserved pair, and θ is calculated from Equation (10). It can be seen that two different thresholds, *e*_1_ and *e*_2_, are set to filter the dynamic features in the SGC algorithm because the features in the bounding box area of an object labelled as dynamic are most likely truly dynamic features. Hence, the features in the bounding box area are more strictly filtered, which means that *e*_1_ is smaller than *e*_2_. Finally, the rest of the static feature pairs are reserved in the *Kps* sequence.

### 3.4. Static Point Cloud Map Creation

Environmental map construction is the core function of the SLAM system. In a static environment, a map is generally constructed by superimposing external environmental information. However, this method may cause some problems in dynamic environments. For example, a constructed map of a dynamic environment will contain motion ghosting caused by dynamic objects, meaning that this method cannot be used in real applications. Thus, a novel static map construction method is introduced in this section.

SGC-VSLAM adds a 3D point cloud map construction module based on ORB-SLAM2 in a new thread. The point cloud map can be constructed by accumulating the 3D points of frames. In the process of map construction, for YOLO v3, the number of the frames participating in map construction must be reduced to ensure that only a few frames are used to obtain a comprehensive environmental map; on the other hand, the dynamic and static information of the frames needs to be screened to ensure that the large dynamic range of the drawing key frames does not participate in map construction. Therefore, SGC-VSLAM selects parts of the key frames to the construct map. The key frame generation mechanism of SGC-VSLAM is the same as that of ORB-SLAM2 and will not be repeated here. The selected key frames for map construction are called drawing key frames. Therefore, the key to building a static map is selecting suitable drawing key frames.

The drawing key frame selection strategy adopted by SGC-VSLAM more strongly depends on the information of the map points observed by the key frames. In detail, a database called *Tmp* is constructed, and all the map points observed by the selected drawing key frames are added to *Tmp*. When the ith new key frame is generated, the map points observed by the key frame *mp* (*i*) and the map points in *Tmp* are compared. If most of the map points in *mp* (*i*) are in *Tmp*, then this frame is filtered out; otherwise, it is selected as the drawing key frame. Thus, the observed map points are used to determine whether the information observed at a key frame is redundant. If the information observed in this key frame is highly coincident with that of the selected drawing key frames, it is no longer selected as the drawing key frame. The detailed drawing key frame selection strategy is expressed in Algorithm 3:
**Algorithm 3.** Drawing key frame selection strategy Input: key frames sequenceOutput: drawing key frames sequence *Tmp*1: for every key frame *i* do2:  if *mp*(i)∩*Tmp* > ε∙*sTmp* do3:    append *mp(i)* to *Tmp*4:   end if5: end for
where *mp*(*i*) is all of the map points observed by the *i* key frame, *Tmp* represents the total map points observed by all of the drawing key frames, and the numbers of map points in *Tmp* are denoted as *sTmp*.

When a drawing key frame is newly generated, its depth image is also reserved, and the mapping module starts to reconstruct the three-dimensional information of the frame. First, The YOLO detection boxes are used to filter out the dynamic pixels, and the depth values of pixels are calculated from the depth image. The pixels of this frame that are dynamic or cannot acquire the depth value from their depth image are filtered out. Then, the spatial position of the rest of the pixels can be calculated by the following equation:(11)$$\left[\begin{array}{l} X_{i}\\ Y_{i}\\ Z_{i} \end{array}\right]=\frac{s_{i}T}{k}\left[\begin{array}{l} u_{i}\\ v_{i}\\ 1 \end{array}\right]$$
where $X_{i}$, $Y_{i}$, and $Z_{i}$ represent the spatial coordinate values of pixel *i*; and $u_{i}$ and $v_{i}$ are the pixel coordinate values; *k* is the parameter matrices of the camera; $s_{i}$ denotes the depth value of pixel *i*, which can be obtained from the depth value; and *T* is the pose matrix of the camera, which is calculated from a visual odometer. Finally, the static point cloud information of all drawing key frames is superimposed to complete the static map construction.

### 3.5. Experimental Setup

To verify the effectiveness and robustness of the proposed algorithm, two experimental tests were performed: A SGC-VSLAM system performance test and a static point cloud map creation test.

The public TUM dataset [36] was used to test the performance of the algorithm. This dataset contains color and depth images featuring scale and rotation changes, with a resolution of 640 × 480. These data are intended to evaluate the accuracy and robustness of the SLAM algorithm in indoor scenes with fast-moving dynamic objects. The real trajectory is obtained by a motion capture system, which consists of 8 high-speed (100-Hz) cameras to ensure the reliability of the real trajectory. The dataset contains four typical types of camera self-motion (halfsphere, rpy, static, and xyz) in the sitting and walking sequences, where halfsphere indicates that the camera moves along the hemisphere trajectory, rpy indicates that the camera rotates along the roll-pitch-deflection axis, static indicates that the camera is basically fixed in place, and xyz indicates that the camera moves along the xyz axis.

All experiments were run on a computer with a 3.40 GHz Intel (R) i7-6700 CPU, 20 GB of memory, an NVIDIA p106-100 GPU, and the Ubuntu 14.04 operating system. The development language was C++. All tests were performed 5 times, and the average value was taken.

## 4. Experimental Results

Two experiments were conducted: The SGC-VSLAM system performance test and the static point cloud creation test. The purpose of the first test was to verify the position accuracy and the stability of the SLAM system in a dynamic indoor environment, and the second experiment was done to test the competency of static map creation when the scene contains independent dynamic objects.

### 4.1. SGC-VSLAM System Performance Test

In this section, the proposed approach was fused to the ORB-SLAM front end as a preprocessing stage. Two aspects of the SGC-VSLAM performance were tested: The positioning accuracy and the operational efficiency of the SLAM system.

#### 4.1.1. Positioning Accuracy Test

The quantitative evaluation was performed using the absolute trajectory error (ATE) and relative pose error (RPE). Table 1, Table 2 and Table 3 list the results of the quantitative test. In Table 1, Table 2 and Table 3, SGC-VSLAM (IQ) (Improved quadtree-based algorithm, IQ), which represents the improved quadtree-based method, was fused to ORB-SLAM2. SGC-VSLAM, the improved quadtree-based method, and the SGC method were all combined into ORB-SLAM2. IMPROVEMENT represents the percentage improvement of the test system over the standard system.

The value of the root mean square error (RMSE) in the table is highlighted as the value of the RMSE that is susceptible to large or occasional errors [37]. Therefore, the robustness of the system can be better represented by the RMSE than by the mean and median. The standard deviation (STD) value is also highlighted because it represents the stability of the system. In the sitting sequence, the dynamic target included only two people in the scene and featured only local limb movement, which belonged to a low dynamic motion sequence labeled with the symbol “*” in tables. For convenience, the TUM dataset fr3_walking_halfsphere is denoted as w_half, and the recording form of the rest of the data is similar.

The test results of the absolute trajectory errors are shown in Table 1, and the positing trajectory of each system on the two test sequences is shown in Figure 5 and Figure 6. It can be seen from Table 1 that SCG-VSLAM(IQ), in terms of its ATE, after integrating the improved quadtree-based algorithm with ORB-SLAM2, can effectively improve the performance of the original system; the RMSE and STD of the SGC-VSLAM system improvement values can reach up to 98.29% and 98.39% in a high-dynamic sequence, respectively. The results indicate that SGC-VSLAM can effectively improve the positioning accuracy and stability of the ORB-SLAM system in a high-dynamic environment. However, the improvements in the RMSE and STD values are not apparent in the low-dynamic sequence because ORB-SLAM2 can filter dynamic features well via its local map mechanism and already has high accuracy; thus, the room for improvement is limited.

Tracking is usually lost in the w_rpy sequence during the ORB-SLAM performance test. This sequence contains not only dynamic objects but also the camera’s rotation and translation, which generate a large number of images with motion blur. Therefore, many low-quality features extracted by the quadtree-based approach may cause the SLAM system to become unstable. Conversely, SGC-VSLAM is relatively stable. It can be seen that SGC-VSLAM in w_rpy and w_half did not improve ORB-SLAM2 as successfully as it did in other sequences. These two sequences had a large number of frames featuring dynamic objects that occupied half of the frame. We found, through experiments, that when dynamic objects occupy half of the current frame, especially when the dynamic objects are only on one side of the image, the error of the epipolar constraint will increase. This phenomenon primarily occurs because SGC-VSLAM first filters the features extracted from the objects labelled as dynamic and then uses the remaining features to calculate the fundamental matrix. When the sampling points of the fundamental matrix are calculated to focus on the local area of the image, the matrix cannot accurately reflect the global information of the whole frame [38].

SGC-VSLAM was also compared with state-of-the-art SLAM systems in a dynamic environment. DynaSLAM [15], DS-SLAM [29], and Detect-SLAM [28] were adopted for these comparisons. These are all excellent VSLAM models that were proposed in recent years. The above systems are all built upon ORB-SLAM2 and adopt the RMSE of the ATE as a quantitative metric. However, the results of ORB-SLAM2 in the same sequence are different between our study and past work. This may be due to the differences in experimental conditions. Thus, the relative accuracy improvement of each system compared to ORB-SLAM2 was adopted as the evaluation metric. The relative accuracy improvement is shown in Table 4.

In terms of the relative accuracy improvement, SGC-VSLAM was only lower than DynaSLAM for the s_half sequence. This is largely because the proposed SGC method can effectively filter the features extracted for dynamic objects.

DynaSLAM is better than SGC-VSLAM in the s_static sequence because it tends to filter the features. DynaSLAM will filter all the features detected on the identified dynamic objects. However, the dynamic features in SGC-VSLAM are judged by their distance from the epipolar line, but some dynamic features with low-dynamic motion states are not detected due to their unsuitable thresholds. In general, SGC-VSLAM offers excellent positioning performance for the tested high-moving sequence compared to the other three SLAM systems.

#### 4.1.2. Real-Time Evaluation

Real-time performance is an essential factor of the SLAM system for practical application. Real-time processing means that the frame rate the algorithm can process is greater than or equal to the frame rate of the input sample; that is, the algorithm can handle the current frame before the arrival of the next frame. The TUM dataset uses a Kinect camera for image acquisition, whose frame rate is 30 FPS (Frames per Second).

Table 5 provides the operation efficiency test results of some major modules. The “(O)” in the table means that the module belongs to ORB-SLAM2. It can be seen that the ORB feature extraction module in SGC-VSLAM was faster than that of ORB-SLAM2. The average time in the main thread of SGC-VSLAM to process each frame was 75.5 ms, including semantic segmentation, dynamic feature filtering, visual odometry estimation, and point cloud map construction. Real-time processing could not be achieved in the current test environment, primarily because the semantic detection module required a large amount of time. YOLO v3 can be used in real-time under better hardware environments, so the real-time performance of SGC-VSLAM can be further improved as the test hardware improves.

### 4.2. Static Point Cloud Construction Test

The proposed novel drawing key frame selection strategy was tested in this section. The results are shown in Figure 7. For convenience, the map created by the key frame generated from ORB-SLAM2 is denoted as MKF, and the map constructed by the drawing key frame is represented as MDKF. It can be seen from Figure 7 that the MDKF had better readability compared to the MKF in the tested sequence, which was influenced insignificantly by the dynamic objects.

In Figure 7c, a significantly incorrect stack of point clouds occurred in the map, and the dynamic objects made the map unreadable. There are two possible reasons for this problem. On the one hand, the accuracy of the camera pose estimation was low and was influenced by the dynamic objects. On the other hand, the key frame included more redundant information, and MKF did not filter the dynamic point clouds, which caused the map to include many stacked dynamic objects. Conversely, the MDKF did not have these problems, which sufficiently proves the effectiveness of the drawing key frame selection strategy. In Figure 7k, the dynamic objects that appeared in the map may have been due to the objects not being detected by YOLO or the objects remaining static between two adjacent frames. The map in Figure 7l missed part of the static scene, primarily because SGC-VSLAM could only use the detection box to filter dynamic pixels. Consequently, some static pixels in the detection box were not constructed in the map and will instead be created when the camera moves away.

### 4.3. Discussion

SGC-VSLAM is designed to improve the robustness of ORB-SLAM2 in a dynamic environment. This model adds dynamic feature filtering and point cloud map creation modules to the RGB mode of ORB-SLAM2.

The result of the experiment shows that the improved quadtree-based method can effectively improve the positioning accuracy of ORB-SLAM2 in a dynamic scenario, which solves the over-uniform distribution of the features extracted from the quadtree-based algorithm [5]. In terms of efficiency, the processing time of each frame in the improved quadtree-based algorithm was reduced about 0.78 ms compared to the quadtree-based algorithm. Moreover, the extraction threshold of FAST was set according to the engineering experience value in the quadtree-based algorithm, which was unable to agree better with real environments. The improved quadtree-based algorithm adopted an adaptive threshold, which can better adapt to different environments.

Many state-of-the-art SLAM systems have made some improvements in detecting dynamic features. For example, DS-SLAM [30], DynaSLAM [15], and PSPNet-SLAM [31] combine deep learning and geometric methods to filter dynamic features. In DS-SLAM, the dynamic features detected by the epipolar constraint method are used to identify dynamic objects. Then, all the features extracted from the dynamic objects are filtered. The SGC method proposed in this paper also uses the epipolar constraint method to detect dynamic features. Unlike the method used for DS-SLAM, the calculated fundamental matrix is based on the features out of the semantic bounding box of objects labelled as dynamic, which can effectively reduce the errors of the calculated fundamental matrix. Moreover, the final filtered dynamic features depend on factors other than semantic information, which reduced the impact of segmentation accuracy on the filtering effect. DynaSLAM features excellent performance when using the TUM dataset, in which the features, regardless of their semantic dynamics and geometric dynamics, are all filtered. However, the Mask R-CNN [26] used in DynaSLAM does not offer real-time performance, which limits its practical applications. PSPNet-SLAM uses a real-time PSPNet network to detect objects labelled as dynamic, and the features extracted from these objects are all filtered out. Similarly, SGC-VSLAM also detects objects labelled as dynamic in a new thread. However, unlike PSPNet-SLAM, SGC-VSLAM will further judge if these objects labelled as dynamic are static, thereby avoiding tracking failures due to insufficient feature points in extreme cases, such as the objects labelled as dynamic occupying over half of the image. Unlike the existing system integrating YOLO and ORB-SLAM2, such as [27,28], the filtering of dynamic features in SGC-VSLAM is not limited to the detection boxes, that is, the static feature points in the detection boxes will also be retained. That can largely solve the problem of tracking failure caused by lack of feature points under some extreme conditions. In terms of positioning accuracy, the state-of-the-art SLAM system (DS-SLAM, DynaSLAM, and Detect-SLAM [29]) were compared with SGC-VSLAM. The results show that SGC-VSLAM offers good positioning accuracy in a high-dynamic environment (the highest improvement value reached 98.29%). However, the improvement value in a slow-dynamic environment was shown to be lower than that of DynaSLAM.

In terms of map construction, there are many excellent methods to construct a static map in a dynamic environment. DS-SLAM can generate an octotree map, and DynaSLAM can create a dense static map using the background inpainting method. These methods all use semantic information to prevent dynamic objects from appearing on the map. However, the drawing frames containing redundant information are not handled well in the above methods. Hence, a novel drawing key frame selection strategy was adopted in SGC-VSLAM. The experimental results show that this method can effectively reduce the redundant information transmitted to the map creator and thereby improve real-time preference.

There is still ongoing research into SGC-VSLAM, which could be improved in three ways. First, the dynamic feature point detection in SGC-VSLAM uses only two consecutive frames. Further improvements may be achieved by using more frames, which may provide more redundant information to filter dynamic feature points. Second, the performance of mapping using SGC-VSLAM is limited mainly by the accuracy of the target detection net, while a high-accuracy semantic segmentation net may improve mapping performance. Third, when objects labelled as dynamic occupy half of the current frame, the calculated fundamental matrix cannot accurately reflect the whole frame, causing some errors of the epipolar constraints. Using a multiframe joint to further judge the state of the target labelled as dynamic may fix this problem.

## 5. Conclusions

In this paper, a novel semantic and geometric constraint-based VSLAM (i.e., SGC-VSLAM) was designed to overcome the degeneration of systems in high dynamic environments. SGC-VSLAM modifies the ORB feature extractor of ORB-SLAM2 and adds two modules: Dynamic feature detection and static map construction. SGC-VSLAM was tested with challenging dynamic sequences from the TUM RGB-D dataset. The following conclusions were obtained through experimental verification:(1)The proposed improved quadtree-based algorithm can effectively improve the positioning accuracy of ORB-SLAM2 in a high-dynamic environment. Moreover, the running time is reduced compared to the standard quadtree-based method.(2)In the SLAM test, SGC-VSLAM significantly outperformed ORB-SLAM2 in its positioning accuracy and robustness in the tested dynamic sequence, and the average RMSE of the ATE decreased by about 96% compared to the original system in a high-dynamic environment. Moreover, comparisons with three state-of-the-art SLAM systems in dynamic environments showed that SGC-VSLAM offers the highest relative RMSE reduction compared to the ORB-SLAM2.(3)The map creation test showed that SGC-VLSM is capable of constructing a static point cloud map without dynamic objects; this map can be re-used in long-term applications.

## Figures and Tables

**Figure 1 sensors-20-02432-f001:**
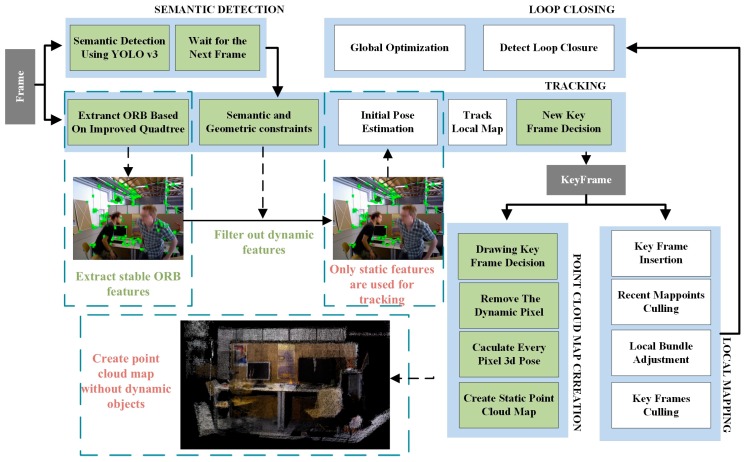
Overall-architecture for SGC-VSLAM. The improved quadtree-based, semantic and geometric constraints’ methods are integrated into the tracking thread, among which only the static features are used to calculate and optimize the camera pose. A semantic detection thread is added to detect objects labelled as dynamic, and a static point cloud map is created from the point cloud map creation thread.

**Figure 2 sensors-20-02432-f002:**
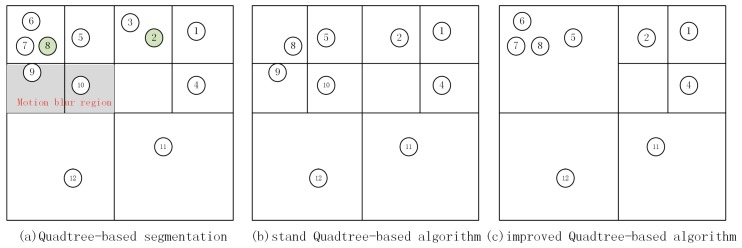
A schematic diagram of the quadtree-based algorithm. The circles denote the feature points. The grey area in (**a**) is the motion blur region, and the green circles represent the features with the highest Harris response values in their nodes. The desired features in (**b**) and (**c**) are set as 9.

**Figure 3 sensors-20-02432-f003:**
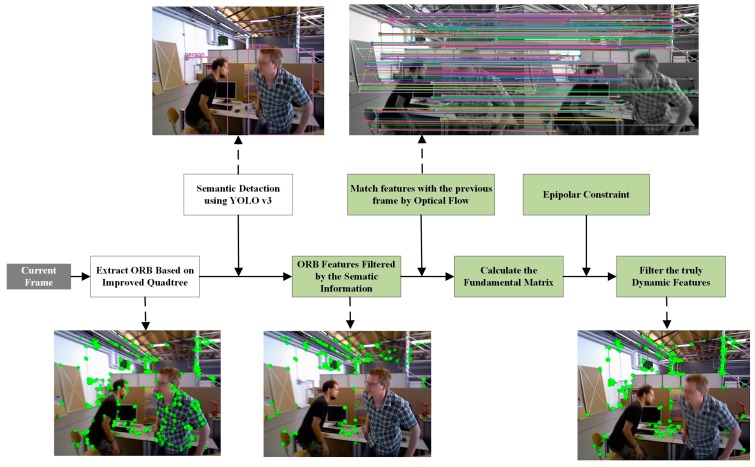
Flow chart of the semantic and geometric constraints’ algorithm. Features extracted from objects labelled as dynamic are first screened by their semantic information. Then, the stable fundamentals are calculated by the remaining features. Finally, all the real dynamic features are filtered out by the epipolar constraint.

**Figure 4 sensors-20-02432-f004:**
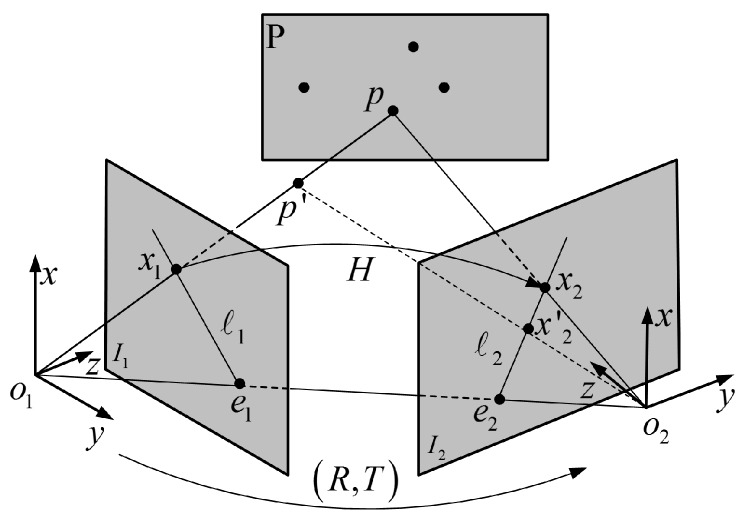
The schematic diagram of the epipolar constraint.

**Figure 5 sensors-20-02432-f005:**
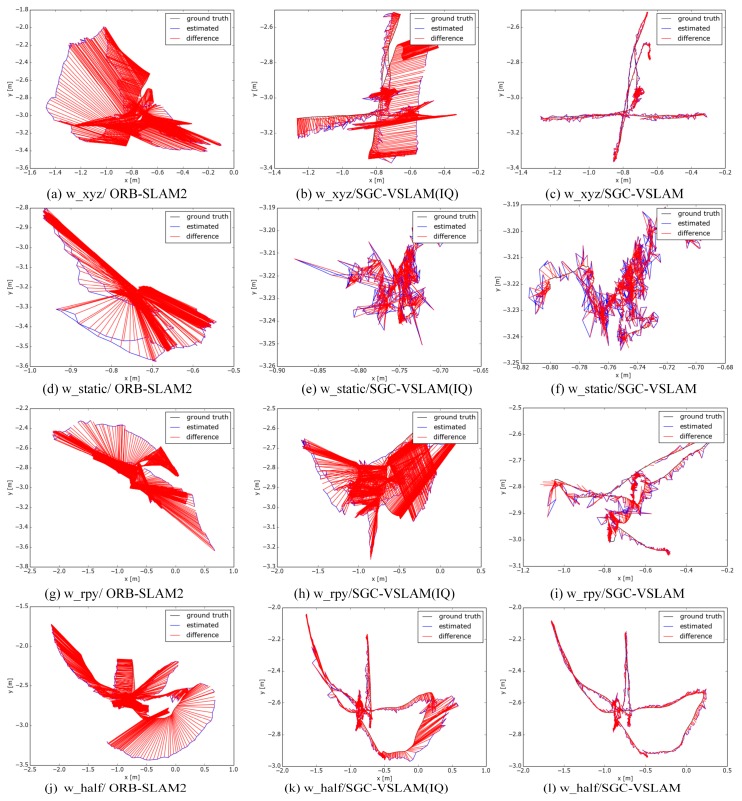
The SLAM trajectory in the high-dynamic sequence. Three systems were tested on the high dynamic sequences. The same row in the figure is the test results of the systems on the same test sequence. The first column in the figure represents the test results of ORB-SLAM2, the second column denotes SGC-VSLAM(IQ) and the third column is the test trajectory of SGC-VSLAM.

**Figure 6 sensors-20-02432-f006:**
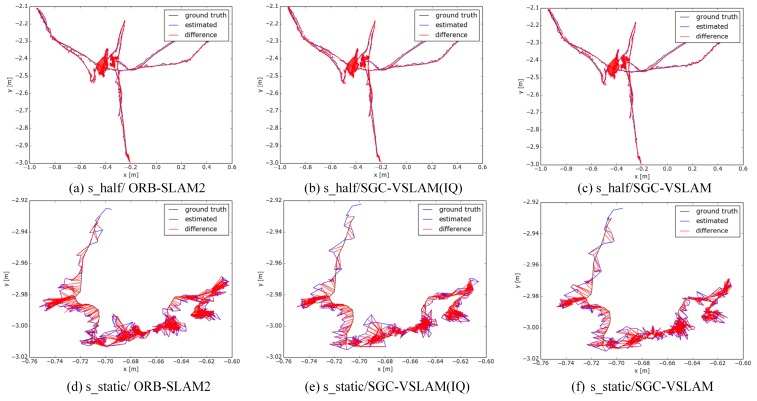
The SLAM trajectory in the low-dynamic sequence. Three systems were tested on the low dynamic sequences. The same row in the figure is the test results of the systems on the same test sequence. The first column in the figure represents the test results of ORB-SLAM2, the second column denotes SGC-VSLAM (IQ) and the third column is the test trajectory of SGC-VSLAM.

**Figure 7 sensors-20-02432-f007:**
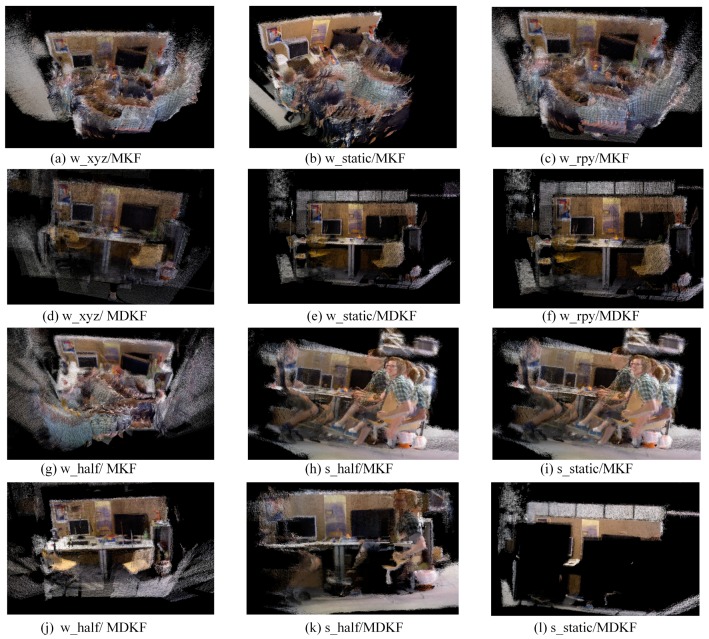
Results of the point cloud map creation. The two map construction strategies were tested on experimental sequences. (**a**–**c**) and (**g**–**i**) are the test results of MKF. Maps built by MDKF and filtered out dynamic objects are shown in (**d**–**f**) and (**j**–**l**).

**Table 1 sensors-20-02432-t001:** Experimental results of the absolute trajectory error.

Sequences	ORB-SLAM2		Improved SLAM System							
			SGC-VSLAM(IQ)		IMPROVEMENT		SGC-VSLAM		IMPROVEMENT	
	rmse	std	rmse	std	rmse	std	rmse	std	rmse	std
w_xyz	0.7787	0.4447	0.1781	0.1354	77.12	69.55	0.0155	0.0072	98.01	98.38
w_static	0.4102	0.1948	0.0116	0.0082	97.17	95.80	0.007	0.0031	98.29	98.39
w_rpy	0.7126	0.2731	0.4373	0.1911	38.63	30.04	0.0278	0.0177	96.10	93.51
w_half	0.4408	0.1878	0.1053	0.0862	76.10	54.07	0.0257	0.013	94.16	93.09
s_half *	0.0174	0.0106	0.0164	0.0101	6.04	5.30	0.0161	0.0101	7.61	5.25
s_static *	0.0098	0.0042	0.0076	0.0036	21.98	15.60	0.0071	0.0035	27.24	16.76

* represents low dynamic sequence.

**Table 2 sensors-20-02432-t002:** Experimental results of the relative displacement trajectory error.

Sequences	ORB-SLAM2		Improved SLAM System							
			SGC-VSLAM(IQ)		IMPROVEMENT		SGC-VSLAM		IMPROVEMENT	
	rmse	std	rmse	std	rmse	std	rmse	std	rmse	std
w_xyz	1.1812	0.6880	0.2484	0.1807	78.97	73.73	0.0222	0.0099	98.12	98.56
w_static	0.5819	0.3783	0.0169	0.0112	97.10	97.05	0.0107	0.0050	98.15	98.69
w_rpy	1.0238	0.5399	0.6167	0.3392	39.77	37.17	0.0404	0.0237	96.05	95.62
w_half	0.6711	0.3794	0.1460	0.1184	78.25	68.79	0.0365	0.0167	94.56	95.60
s_half *	0.0256	0.0153	0.0240	0.0146	6.37	4.45	0.0233	0.0143	8.82	6.49
s_static *	0.0151	0.0073	0.0116	0.0056	22.69	22.18	0.0115	0.0055	23.65	23.55

* represents low dynamic sequence.

**Table 3 sensors-20-02432-t003:** Experimental results of the relative rotational trajectory error.

Sequences	ORB-SLAM2		Improved SLAM System							
			SGC-VSLAM(IQ)		IMPROVEMENT		SGC-VSLAM		IMPROVEMENT	
	rmse	std	rmse	std	rmse	std	rmse	std	rmse	std
w_xyz	22.1140	12.5038	4.1150	3.2911	81.39	73.68	0.6252	0.3739	**97.17**	**97.01**
w_static	10.4395	6.9241	0.3610	0.2006	96.54	97.10	0.3000	0.1320	**97.13**	**98.09**
w_rpy	18.4619	10.6122	11.3437	6.6349	38.56	37.48	0.9047	0.4969	**95.10**	**95.32**
w_half	17.0926	9.5850	2.9947	2.4663	82.48	74.27	0.8338	0.3601	**95.12**	**96.24**
s_half *	0.7147	0.2998	0.6647	0.2821	**6.99**	**5.93**	0.6817	0.2878	4.61	4.01
s_static *	0.3934	0.1702	0.3458	0.1492	12.10	12.36	0.3374	0.1478	**14.22**	**13.19**

* represents low dynamic sequence.

**Table 4 sensors-20-02432-t004:** Comparisons of the relative RMSE [m] improvement for SGC-VSLAM against the state-of-the-art system for the dynamic sequence of the TUM RGB-D dataset.

Sequence	DS-SLAM	DynaSLAM	Detect-SLAM	SGC-VSLAM
w_xyz	96.71%	96.73%	97.62%	**98.01%**
w_static	97.91%	93.33%	—	**98.29%**
w_rpy	48.97%	94.71%	—	**96.10%**
w_half	93.76%	92.88%	66.94%	**94.16%**
s_half	—	**15%**	-27.62	7.61%
s_static	25.94%	—	—	**27.24%**

**Table 5 sensors-20-02432-t005:** Results of the operational efficiency test.

Module	ORB Feature Extraction (O)	ORB Feature Extraction	Dynamic Feature Filtering	Semantic Detection
Thread	Tracking	Tracking	Tracking	Semantic detection
Time (ms)	9.07	8.29	38.79	55.47

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
