# Peer review of "SGC-VSLAM: A Semantic and Geometric Constraints VSLAM for Dynamic Indoor Environments"

_sensors, 2020, doi:10.3390/s20082432_

Round 1
Reviewer 1 Report
In this paper is presented an approach for Visual Simultaneous Localization and Mapping (VSLAM) that integrates semantic and geometric constrains to deal with real world objects dynamics.
The proposed approach lies on previously well-known algorithms, particularly the ORB-SLAM2 method.
Authors propose the improvement of a variant for Quadtree-based feature detection method; in which, better features are selected.
Been important the proposed improvement, but not considered as the main contribution, authors should clarify better their improvement as is not clear why dynamic objects should generate inferior Harris values.
The proposed methodology approach lies mainly on the consideration of a sequential phases of a semantic segmentation, followed by a particular combination with geometrical constrains to effectively erase feature points on dynamic objects.
The steps can be summarize as follow: a) detect Quad-tree ORB improved feature points, b) detect semantic objects with YOLO v3 in the scene, c) without considering the feature points belonging to the bounding boxes of the objects detected compute the fundamental matrix due to motion, and finally d) filter feature points inside and outside bounding boxes with different criteria according to the geometry constrains.
Authors presents evaluations and comparisons with previous approaches, using a very well-known database, the public TUM database, showing better results for the majority of the experiments.
While the following paper is very new, authors should discuss main differences (particularly on semantic, as geometric constraints are not considered) with the following paper.
Han and Z. Xi, "Dynamic Scene Semantics SLAM Based on Semantic Segmentation," in IEEE Access, vol.8, pp. 43563-43570, 2020.
In spite of a marginal contribution, the paper is complete. However, it requires a lot of work to arrive to an acceptable version. The English writing should be radically improved, as in the actual state it is very difficult to follow some sections, it seems that authors have reviewed differently the writing of their paragraphs, making a very deficient document.
Below are given some examples, of corrections to do, however are not exhaustive and the whole document should be reviewed. It is recommended to use some editorial service for that.
There are many mistakes on Figure 1, “sematic”, “improbed”, “detectin” and should be written “local mapping” and not “loop mapping”
The Feature Quadtree-based algorithm on page 4 should be emphasized with a different format from the used for regular text.
In page 4 in line 156, it should be written “circles 9 and 10 are dynamic features” and not “circle points 9 and 10 is the dynamic features”
On same page, it Is not clear, why? the circles 9 and 10 being dynamic features require to have low Harris values, please clarify. Also, it is required to explain on this paragraph, what it’s write on Fig.2 caption (circles with low Harris values) to clarify.
On line 58 replace “is” with “are”.
The use, on line 60, of the word “positing”, seems to be a mistake.
On line 162 it should be written “has” and not “have”.
Please rewrite the complete paragraph, from line 153 to 163, to be clearer and please, verify the complete coherence.
On page 5
All terms and superscripts on equation (1) should be described.
Correct the word “segmeted” on line 183.
Sentence on line 191 is unclear, please rewrite it.
“First, features in the node with only one feature are reserved as the final feature”
The same for sentence on lines 191 to 192. The multiple use of word "feature" should be avoided; for example, with the use of pronouns.
On lines 194 and 200 the plural-singular concordance should be reviewed.
Correct the use Fig2. on line 197.
On line 199, the use of word “selected” should be preferred to the used word “reserved”, as well as, on many other places along the paper.
Avoid the use of Figures for pseudo-codes (e.g. Figure 3, 6 and 7), it's better to use tables for it.
Standardize references to figures, for example in line 201-202, where has not been used the common “Fig.3”.
On Algorithm 1, verify line 5;
The use of italics, for variables and parameters, when writing them on paragraphs should be used, for example on lines 205-207.
In line 221 should be written “PASCAL” and not “PASCSAL”
At the end of this section, it’s not clear the relation of the proposed approach, with dynamic features mentioned before, please clarify.
In page 7, line 29 replace “little” by “few”, and “own parameters” by “internal parameters”
Verify lines 279, 280, 281, and 282 as they present the same type of the mentioned mistakes.
It is better the review of whole document by a native English speaker before sending to an editorial editing service.
And finally, please verify format and particularly the spaces between words in the section of references.
Author Response
Thank you for your comments on our manuscript entitled “SGC-VSLAM: A Semantic and Geometric constraint VSLAM for indoor Dynamic Environments” (ID: 746977). Those comments are constructive for revising and improving our paper, as well as the essential guiding significance to other research. We have studied your comment carefully and made corrections which we hope meet with approval. We have uploaded the responses to your comments, please see the attachment.

Reviewer 2 Report
This research proposed the SGC-VSLAM to improve the positioning precision and the robustness of the robots in indoor dynamic environments, and its performance is verified by experiments in positioning accuracy, real-time evaluation and static point cloud creation. Some of the issues should be mentioned:
- Line 156: ‘the circle point 9 and 10 is the dynamic features and have the low Harris response value’. In the quadtree algorithm, if 9 and 10 get the lower Harris response value, they will be removed from the features. But they are still there in Fig.2 (b). This sentence is irrelevant to what the author is trying to express and may even cause confusion.
- Line 249: ‘p2=Rp2+t’, the second p2 should be p1.
- Line 271: The threshold is very important to filter the dynamic features, but the author does not mention how to determine the value, such as function or references. This should be provided with more details to prove the adaptive of this parameter.
- Table 4: The time listed in the table is only a part of the SLAM system, so does the time of the whole system meet the real-time requirement, such as FPS and other parameters. The improvement of the QDtree reduces the extraction of unnecessary feature points and optimizes the time of feature extraction. But it also increases the operation time of YOLO algorithm and dynamic feature filtering. Therefore, some explanations are necessary.
- ‘table’, ‘figure’, ‘section’ and so on should be written with first letter in captain format.
- Line 490, ‘can re-used’ is wrong in English style.
- ‘And’ in line31 should be ‘and’.
- The comparisons of RMSE between your research and other algorithms are based on different data, how can you prove your proposed method is better than theirs? Or they are based on the same real-time system?
Author Response

(The authors gave the same response as above.)

Reviewer 3 Report
This article presents a SLAM in dynamic environment that relies on a improved quad-tree algorithm and on using YOLOv3 to filter out dynamic objects.
As it is the article is too poorly written to be considered for publications. Also numerous other publication have combined ORB2-SLAM with YOLOv3 for filtering objects (such as [1]), they should be presented in the introduction, and it should be made clear how this article differ from them. Usage of geometry constraints, such as epipolar geometry, have been part of SLAM algorithms for more than 20 years. The only originality in this paper would come from the improved quad-tree algorithm, but due to the low quality of English, I failed to understand what the authors were doing.
Otherwise, the work is correctly evaluated and compared to state of the art.
* English language and style
Overall, the article is extremely poorly written, which makes it very very hard to understand. It needs serious editing before I can make a proper assessment of the work. And I am not talking about spelling mistakes, but poorly constructed sentences.
For instance, l188-190
"The last step is that reserving the final features from every child node, which is the most
challenge task for improved algorithm. The feature retention has a hard decision when large numbers of node with more than one features occurs due to the limited iteration times."
If I understood it correctly, should be written as:
"The most challenging task of the improved algorithm is the last step which is the selection of the final features of every child nodes. The limited iteration time makes the feature selection a hard problem with a large numbers of nodes."
And this is just an example, nearly all sentences need to be corrected. And please, make sure you use the same name for the algorithm in all places MGC-VSLAM vs SGC-VSLAM.
"Study on Slam Algorithm Based on Object Detection in Dynamic Scene" Ping Li ; Guoqing Zhang ; Jianluo Zhou ; Ruolong Yao ; Xuexi Zhang ; Jianluo Zhou
V https://ieeexplore.ieee.org/document/8861669
Author Response

(The authors gave the same response as above.)

Reviewer 4 Report
This paper presents a Visual Simultaneous Localization and Mapping (VSLAM) system that uses an object detection network to detect features belonging to dynamic objects, and remove them in combination with geometric constrains.
Although the idea is nice, the paper needs so much "love". It has to be reworded. The use of English needs deeper revision too. For example, I list some mistakes just in the first 50 lines:
15 - , this paper present(s) SGC-VSLAM
16 - building on the RGB-D mode of the ORB-SLAM2 <- doesn't sound good
20 - priori -> a priori or prior
23 - a static point cloud is estimated (by used of)(and used by) the proposed new key frame selection strategy
31 - Simultaneous Localization and Mapping or simultaneous localization and mapping.
41 - gathers in strong texture regions
42 - large number of dynamic features will be extracted (when)(with) dynamic objects with strong texture information
45 - the vast majority of SLAM system(s)
46 - static computation model may (be) impacted by (the) dynamic objects in the environment.
47 - in (a) dynamic environment, or in dynamic environment(s)
48 - which limits (the intelligent) SLAM system(s) (using)(used) in populated real-world scenario over a long period of time.
49 - on the SLAM system is a critical challenge.
52 - Aiming to the problem of the ORB features are prone to gathered together on the strong texture information region <- this sentence doesn't make sense
This seriously hampers the understanding of the concepts being described.
Some other points:
- The intro of section 2 is not understandable.
- Figures have a low quality. Please use vector formats.
+ Fig.1 -> On Impro(v)ed quadtree
+ Fig.4 is confusing.
- Section 2.2 is a mesh of concepts and descriptions.
- Please insert Algorithms as Algorithms, not as images.
- Yolo doesn't provide masks, but bounding boxes.
- You say that: "When the detection box occupies more than half of the current frame, the rest features are little, which may cause the failure of the tracking thread in the SLAM system.". I guess that could also hamper the computation of the Fundamental matrix. This should be analyzed in the evaluation section.
- In 2.4, why do you use YOLO for filtering the dynamic features? Why not use the output of 2.3.3?
- The novel drawing key frame selection strategy adopted by the SGC-VSLAM is not properly described.
- The role of depth images in your pipeline is not clear.
- In table I, for example, that improvement in some of the sequences (over 98%) indicates that the comparison is not fair. You should fuse that section with 3.1.3.
- The real time evaluation done is poor. First of all, what is real time? you have to define it. You say that the dynamic filtering module spends about 38.79ms, the semantic detection takes 55.47, and the ORB feature extraction 8.29. That sums up +100ms. Without taking into account the rest computations, that limits the operation to 10Hz.
Summarizing, the paper needs:
- A serious and deep revision of English.
- A better description of the system proposed.
- Upgrade the images quality.
- Improve the evaluation section, mainly the part regarding real time evaluation, and including some extra analysis like the effect of big bounding boxes provided by YOLO.
Author Response

(The authors gave the same response as above.)

Round 2
Reviewer 1 Report
Authors have made significant corrections to the whole paper, including errors correction both grammatical and orthographic.
They have included the mentioned reference and attended all suggestions.
Paper now can be accepted.
Author Response
Dear reviewer,
Thank you very much for your help in revising our manuscript, your valuable comments and suggestions are constructive for revising and improving our paper, as well as the essential guiding significance to other research.
Thank you and best regards.
Yours sincerely,
Guohao Fan
Reviewer 2 Report
The revision version of this manuscript improved a lot when it compared with the original version. Hence, I recommended acceptance for current version according to my knowledge.
Author Response

(The authors gave the same response as above.)

Reviewer 3 Report
I would suggest to include a contribution list at the end of the introduction. What you gave in response 2 was good information for me.
Regarding the selection of points in the QuadTree, do you use any information from yolo? Or how do you know that the points you selects are more dynamic? Is it just motion blur? But if you have motion blur, don't you get lower Harris response? On l208 you say that you keep the point with highest response, which I would assume has higher sharpness and more likely to be static.
Regarding epipolar geometry, you say that you use it to detect point that are dynamic (l295) but classified by yolo as static, have you thought of also doing it the other way around, if a point is wrongly classified as dynamic, can you use epipolar geometry to classify it as static? (this can happen for many reasons, mainly because the bounding box from yolo cover more than the actual object)
Suggestion of improvement:
- I am not sure "a priori" has the meaning you intend, I would suggest to use "objects labelled as dynamic", then it is clearer that you mean that it is the result of classification state
- l15 "presents SGC-VSLAM" (no an)
- l235 reformulate something like "YOLO v3 can be implemented in real time on embedded platform"
- l249 maybe something like "SGC-VSLAM filters features using geometric constraints..."
- l532 improve -> fix
Author Response
Dear reviewer,
Thank you very much for your comments and valuable advice. We have studied the valuable comments from you and tried our best to revise the manuscript. The point to point response to your comments that we have uploaded by attachment. Please see the attachment.
Yours sincerely,
Guohao Fan

Reviewer 4 Report
The authors did a good work reviewing the paper, specially the use of English, although as commented below there is room to improve:
Sentences to review (this is not an exhaustive list):
42 - For example, the feature points extracted by the standard ORB algorithm often gather in the strong texture regions [10].
52 - there remains -> it remains
53 - Because the ORB features are prone (to gather)(to concentrate?) in the strong texture region
63 - of the SLAM system -> of a SLAM system
108 - ORB-SLAM2 in dynamic environment(s)
116 - In this section, SGC-VSLAM will be introduced in detail. This section -> two sentences with the same beginning.
117 - architecture chart -> that expression is not used
121 - the method of building -> the method for
454 - Real-time pos?
The proper naming of Mask is Mask R-CNN. Please fix that in the whole paper.
And two more things that have to be addressed:
- The introduction section is weird: there are so much text about related work (that could be moved to a new 2. Related work section, and just 2 sentences about the proposed system. The paper contribution must be extended.
- The general quality of the images has not been improved. For example, in Figure 1, the reader shouldn't make an effort to understand what the text says. Such text is blurred. The same happens with the rest of figures, and it must be fixed before possible acceptance. If vector formats are not possible (take into account that, for example, pdf files can be optimized), export images with a higher quality please.
Author Response

(The authors gave the same response as above.)
